# Cerebrospinal Fluid Alpha-Synuclein Improves the Differentiation between Dementia with Lewy Bodies and Alzheimer’s Disease in Clinical Practice

**DOI:** 10.3390/ijms232113488

**Published:** 2022-11-04

**Authors:** Matthieu Lilamand, Josué Clery, Agathe Vrillon, François Mouton-Liger, Emmanuel Cognat, Sinead Gaubert, Claire Hourregue, Matthieu Martinet, Julien Dumurgier, Jacques Hugon, Elodie Bouaziz-Amar, Claire Paquet

**Affiliations:** 1Université Paris Cité and Inserm UMRS-1144 Optimisation Thérapeutique en Neuropsychopharmacologie, 75006 Paris, France; 2Cognitive Neurology Center, Lariboisière-Fernand Widal Hospital (GHU AP-HP.Nord), 75010 Paris, France; 3Department of Geriatrics, Lariboisière-Fernand Widal Hospital (GHU AP-HP.Nord), 75010 Paris, France; 4Biochemistery Department, Lariboisière Hospital (GHU AP-HP.Nord), 75010 Paris, France

**Keywords:** alpha-synuclein, Dementia with Lewy Bodies, Alzheimer’s disease, neurodegenerative disorders, biomarker discovery

## Abstract

Background: Alpha-synuclein, abnormally aggregated in Dementia with Lewy Bodies (DLB), could represent a potential biomarker to improve the differentiation between DLB and Alzheimer’s disease (AD). Our main objective was to compare Cerebrospinal Fluid (CSF) alpha-synuclein levels between patients with DLB, AD and Neurological Control (NC) individuals. Methods: In a monocentric retrospective study, we assessed CSF alpha-synuclein concentration with a validated ELISA kit (ADx EUROIMMUN) in patients with DLB, AD and NC from a tertiary memory clinic. Between-group comparisons were performed, and Receiver Operating Characteristic analysis was used to identify the best CSF alpha-synuclein threshold. We examined the associations between CSF alpha-synuclein, other core AD CSF biomarkers and brain MRI characteristics. Results: We included 127 participants (mean age: 69.3 ± 8.1, Men: 41.7%). CSF alpha-synuclein levels were significantly lower in DLB than in AD (1.28 ± 0.52 ng/mL vs. 2.26 ± 0.91 ng/mL, respectively, *p* < 0.001) without differences due to the stage of cognitive impairment. The best alpha-synuclein threshold was characterized by an Area Under the Curve = 0.85, Sensitivity = 82.0% and Specificity = 76.0%. CSF alpha-synuclein was associated with CSF AT(N) biomarkers positivity (*p* < 0.01) but not with hippocampal atrophy or white matter lesions. Conclusion: CSF Alpha-synuclein evaluation could help to early differentiate patients with DLB and AD in association with existing biomarkers.

## 1. Introduction

Over the past ten years, Cerebrospinal Fluid (CSF) biomarkers (β-amyloid (Aβ), total Tau (t-Tau), phosphorylated Tau181 (pTau)) have dramatically improved the accuracy of the molecular diagnosis of Alzheimer’s Disease (AD) in clinical practice. However, several differential diagnoses remain challenging due to the poor specificity of clinical symptoms linked to neurocognitive disorders and sometimes because of the overlap between neurodegenerative disorders, vascular cognitive impairment and brain aging. The diagnosis of Dementia with Lewy Bodies (DLB), the second cause of neurodegenerative cognitive impairment after AD [1], is based on neurological signs associated with core clinical criteria, as described in the fourth consensus report of the DLB Consortium [2]. These criteria have improved the diagnostic sensitivity for DLB; however, detection rates in clinical practice still remain suboptimal, with many cases misdiagnosed as AD or other α-synucleinopathies [3]. Furthermore, at autopsy, mixed Lewy-related and AD pathologies are common and could involve up to 50% of carefully diagnosed subjects classified as AD patients [4].

DLB is neuropathologically characterized by an accumulation in the brain of abnormally aggregated and phosphorylated alpha-synuclein (α-syn), a 140 amino-acids protein, which is mainly expressed in presynaptic terminals [5]. This accumulation is often associated with brain amyloid deposition, and both amyloid and α-syn participate in neurodegeneration [6]. As a key protein in the pathophysiology of DLB, α-syn represents a candidate biomarker to discriminate individuals with DLB from those with AD or Neurological Controls (NC). However, so far, studies examining the potential role of CSF α-syn as a diagnostic biomarker have brought conflicting evidence. The published studies found either no difference in CSF α-syn concentrations between AD and DLB patients [7,8,9,10,11], or increased α-syn levels in DLB [12], but also decreased levels in DLB [13,14,15,16,17,18,19,20,21]. These discrepancies might result from methodological issues, such as inter-assay and inter-laboratory variations in the measurement of CSF α-syn, or inconsistency regarding the diagnostic criteria of the neurocognitive disorders. Moreover, the relationship between the stage of cognitive impairment and α-syn concentrations in patients with AD and DLB has been studied little. 

Our primary objective was to compare the CSF α-syn levels, measured by a validated ELISA kit, in extensively-phenotyped AD, DLB patients or neurological control individuals from a clinical practice cohort of patients recruited in a tertiary memory clinic. The secondary objectives were to study the relationships between the concentration of CSF α-syn and (i) CSF amyloid and Tau biomarkers, (ii) the main neuroimaging characteristics of these individuals, (iii) the Mild Cognitive Impairment (MCI) or dementia stages, in patients with DLB or AD. 

## 2. Results

We retrospectively selected 85 DLB patients and 105 NC, among whom 45 DLB patients and 32 NC individuals met all the inclusion criteria (See Appendix A); we also randomly included 50 patients with AD confirmed by CSF biomarkers. The demographic and clinical characteristics of the population are displayed in Table 1. Overall, we included 127 participants with a mean age of 69.3 (8.1) years, mainly women (58.3%). Most patients with AD and NC were women (respectively: 72.0% and 65.6%), whereas most subjects with DLB were men (62.2%). Patients with DLB and AD were older than NC individuals: 70.1 (7.0) and 71.7 (8.2) years old, respectively, vs. 64.4 (7.7) (*p* < 0.001). Patients with DLB reported hallucinations in 28.9%, fluctuations in 22.2%, parkinsonism (or positive DAT-SCAN imaging) in 60.0% and RBD in 33.3%. There was no difference regarding hippocampal atrophy between individuals with AD and those with DLB (*p* = 0.80).

### 2.1. Comparison of CSF α-syn between Patients with DLB and AD

The mean CSF α-syn levels were significantly lower in the DLB group than in the AD group: 1.28 (0.52) ng/mL vs. 2.26 (0.91) ng/mL, respectively (*p* < 0.001). There was no significant difference between DLB and NC. In addition, α-syn levels were higher in individuals with AD than in NC (2.26 (0.91) ng/mL vs. 1.56 (0.60) ng/mL, *p* < 0.001) (Figure 1).

The mean concentration of CSF total α-syn was significantly increased in patients with AD as compared to patients with DLB or NC (*p* < 0.001). Further analysis did not show any significant differences between early (MCI) and late (dementia) stages in both patients with AD (*p* = 0.62) and in those with DLB (*p* = 0.30) (Figure 2). 

The best threshold of α-syn to discriminate patients with DLB from patients with AD was 1.50 ng/mL. With this value, we obtained a sensitivity (Se) = 82.0%, a specificity (Spe) = 76.0%, and hence, an area under the curve (AUC) = 0.85 (Figure 3).

### 2.2. Relationship between CSF α-syn, CSF AT(N) Biomarkers and Brain MRI Abnormalities

The mean concentration of total CSF α-syn was significantly increased in patients with brain amyloïdopathy (A+), tau pathology (T+) and neurodegenerescence (N+) (Table 2). There was no difference in total CSF α-syn between the various stages of white matter lesions or hippocampal atrophy (Table 2).

## 3. Discussion

In this study, we observed significantly higher concentrations of CSF α-syn in AD patients than in those diagnosed with DLB or in NC individuals. However, there was no significant difference in CSF α-syn concentrations between the DLB and NC groups. CSF α-syn was neither associated with the stage of hippocampal atrophy nor with white matter lesions on brain MRI. Among the DLB or AD groups, there was no difference in CSF α-syn concentrations regarding the stage of cognitive impairment (MCI vs. dementia). 

α-syn has been identified as a toxic protein to human neurons and a trigger to neuronal death [22]. More precisely, Lewy inclusion-bearing neurons were shown to selectively die, whereas nonbearing neurons survived [23]. In this context, the distinctive aspects in the pathophysiology of DLB and AD could explain the difference in CSF α-syn levels between these two neurodegenerative disorders. On the one hand, in DLB, α-syn is misfolded and phosphorylated, leading to intraneuronal inclusions and synaptotoxicity. Several mechanisms explain the synaptotoxicity due to aggregates of α-syn: a decrease of neurotransmitter release resulting from the blockage of the SNARE-mediated docking between presynaptic vesicles and presynaptic membrane [24], a decrease in the pool of readily releasable presynaptic vesicles [25] or a destabilization of microtubules [26]. However, the aggregates of α-syn in Lewy bodies and Lewy neurites do not seem to play a role in the neuronal disturbances [27]. On the other hand, in AD, the mechanism is quantitative and seems to be characterized by an overproduction of wild-type α-syn that may be interpreted in two ways. First, the elevation of CSF α-syn in AD might be the consequence of an important release of α-syn due to synaptic demise by the same mechanism as protein Tau release from degenerating cells. Second, there might be an overexpression of α-syn in the central nervous system of patients with AD representing a defense mechanism to regulate the neuronal homeostasis. Inflammation in the central nervous system is a key feature of AD. This inflammation provokes an increase of reactive oxygen species characteristic of oxidative stress, modifying α-syn such that it has a higher tendency to aggregate [22,28]. Lee et al. observed an overproduction of wild-type α-syn in NT2 (human teratocarcinoma) and SK-N-MC (human neuroblastoma) cell lines after an incubation with H_2_O_2_, associated with delayed cell death, in comparison with control cell lines [29]. Therefore, α-syn might play a defensive role against oxidative stress. However, this mechanism could be deleterious if it persists: several studies showed a synaptotoxicity associated with an overproduction of wild-type α-syn, involving the same mechanisms as the aggregates of misfolded α-syn [25,30,31]. This accumulation could also lead to a disruption of lysosomal activity [32]. As in the study of Bousiges et al., we found that CSF α-syn concentration was higher in MCI due to AD than in MCI due to DLB. Since inflammation occurs early in the pathophysiology of AD [33], this result could support the hypothesis of overproduction of α-syn, at least in response to inflammation in AD. 

As a result, both aggregates of α-syn in DLB and overexpression of wild-type α-syn in AD favor synaptotoxicity. Nevertheless, previous evidence suggested that the destruction of the synapse was more important in AD than in other neurodegenerative diseases [34]. Patients with DLB experience lower neurodegeneration and preserved cholinergic receptors than those with AD [35]. Our findings are also consistent with the hypothesis that CSF α-syn could also reflect synaptic loss being more severe in AD than in DLB patients [34]. 

So far, several studies have examined CSF α-syn in DLB in comparison with AD, other neurodegenerative disorders or NC (for details, see Appendix A). Bousiges et al. [21], who also used another standardized kit (INNOTEST©, Fujirebio Europe, Ghent, Belgium) to measure CSF α-syn, defined a cut-off of 1.39 ng/mL (close to ours: 1.50 ng/mL) for discriminating AD from DLB and obtained an AUC = 0.78, Se = 72.3%, Spe = 76.1%. In three other studies assessing the discriminating value of α-syn between AD and DLB, the accuracy of this biomarker was notably lower than in the former studies or in our study [12,16,36]. Sensitivity ranged between 50.0% and 80.3%, and specificity ranged between 65.0% and 94.4%. The main difference between the three latter studies and the two former was the lack of standardized ELISA kit for α-syn measurement. Moreover, the criteria used for defining AD and DLB were less stringent than in our cohort: third report of the DLB Consortium for DLB criteria and/or absence of CSF biomarkers for AD. Previous reports have underlined that up to 30% of patients diagnosed with AD did not have neuropathological lesions of the disease [37]. Furthermore, some biomarker profiles may question the diagnosis of DLB, even though the patient meets the clinical criteria for probable DLB, for example, in the case of elevated t-Tau and p-Tau, in the CSF. According to our results, CSF total α-syn may meet the international criteria for the diagnosis of neurodegenerative diseases with a good sensitivity (Se = 82%) despite the limited specificity (Sp = 76%). 

The main strength of our study is the analysis of clinical and imaging data from patients seen in clinical routine practice. Our findings may be extended to the broader population of patients from memory clinics, and CSF α-syn may be considered as an additional biomarker in this setting. However, several limitations must be acknowledged. NC individuals were younger than the patients with neurodegenerative conditions. Nevertheless, previous studies comparing the concentration of α-syn between patients with AD and DLB and NC presented similar median age [14,18,20]. We only measured CSF total α-syn concentration. Future studies quantifying a larger range of CSF α-syn species, such as phosphorylated at serine 129 α-syn or soluble α-syn oligomers, would provide additional insights into the role of α-syn as a potential biomarker of LBD. Current new technologies (e.g., Real-Time Quaking-Induced Conversion and Protein Misfolding Cyclic Amplification (PMCA)) are being developed to measure abnormal α-syn, but their usefulness in daily clinical practice of laboratories deserves further investigation [38]. However, validated ELISA assay may easily be implemented in daily laboratory practice. Finally, we lacked neuropathological confirmation of the diagnoses, which could have improved the accuracy of CSF α-syn to discriminate AD from DLB. We cannot exclude the presence of mixed pathology in patients with AD or DLB or brain α-syn deposition due to normal aging.

## 4. Material and Methods

### 4.1. Design

We conducted a monocentric, retrospective, cross-sectional analysis of CSF and clinical characteristics from patients in an expert tertiary memory clinic: the Cognitive Neurology Center (CNC) Paris Nord (Lariboisière-Fernand-Widal Hospital, Assistance Publique-Hôpitaux de Paris, Université de Paris-Cité) France. This department has expertise in the diagnosis and care of patients with cognitive disorders and neurodegenerative diseases.

### 4.2. Participants

BioCogBank is a monocentric clinical practice cohort of 1500+ patients with cognitive complaints prospectively recruited in the CNC since 2008. All patients included in BioCogBank underwent neurological and neuropsychological assessments, neuroimaging and CSF AD biomarkers analysis for cognitive complaints. Plasma and CSF samples have been biobanked for most of these patients. CSF results were classified according to the AT(N) classification [39]. A consensual clinical diagnosis was performed after discussion between a multidisciplinary team of neurologists, neuroradiologists, geriatricians and biochemists, according to the last validated diagnostic criteria (see below). The stage of cognitive impairment in individuals with DLB or AD: MCI or dementia were defined according to the criteria of the Diagnostic and Statistical Manual of Mental Disorders, fifth edition (DSM-V) [40]. 

We retrospectively identified, from participants included in the cohort between 2008 and 2021, patients fulfilling criteria for DLB, taking into account the information recorded during the clinical follow-up [2]. All patients with DLB patients and NC individuals with available CSF samples were included. In parallel, we randomly included the same number of patients with AD meeting the clinical criteria of Mc Khann’s criteria [41] and confirmed by CSF biomarkers (i.e., with CSF A + T+N+ profile). These individuals with AD and available additional CSF samples were included over the same inclusion period and were individually matched for age and gender with the patients with DLB. The patients with DLB having a full A + T+N+ profile (i.e., indicating mixed pathology with AD) were excluded [2]. NC individuals met the following criteria: absence of cognitive decline, normal neurological examination, negative amyloid and Tau CSF biomarker profile (i.e., A-T-Nx), non-extensive white matter lesions (Fazekas score ≤ 2) [42] and absence of mild hippocampal atrophy (Scheltens bilateral score ≤ 2) on brain MRI [43], normal brain PET-scan, DAT-scan or PET-DOPA (when performed). 

### 4.3. Clinical, Biological, Neuroimaging Variables of Interest

The following cardiovascular risk factors were collected: smoking status (including active smoking or having quit < 3 years ago), hypertension (systolic blood pressure ≥ 140 mmHg and/or diastolic blood pressure ≥ 90 mmHg), diabetes (defined by two measures of glycemia after fasting ≥ 1.26 g/L [44]) and dyslipidemia (defined by an increase of the concentration of triglycerides and/or low-density lipoprotein (LDL) cholesterol according to the European Society of Cardiology (ESC)/European Atherosclerosis Society (ESC) guidelines [45]). Significant alcohol consumption was defined according to the World Health Organization (WHO) guidelines (≤3 glass/day for men, ≤2 glass/day for women). 

The following clinical symptoms associated with DLB were recorded: visual hallucinations (reported by the patient or a relative/caregiver), parkinsonism or dopamine denervation (defined by the presence of akinesia, hypertonia, tremor or a presynaptic dopaminergic denervation on imaging data), fluctuating cognition, Rapid eye movement sleep Behavior Disorders (RBD), reported by patients and relatives and/or using a polysomnography when performed. Global cognitive function was assessed by the Mini Mental State Examination (MMSE) score [46]. 

DNA was extracted from peripheral blood samples for ApoE genotyping, and participants were classified as carrier or non-carrier of at least one *APOE* ε4 allele. Several characteristics of brain MRI were collected for the present analysis: hippocampal atrophy using the visual Scheltens score, defined by the mean values between the right and left sides and periventricular white matter hyperintensities (Fazekas score). 

### 4.4. CSF Samples and Analysis

CSF samples were obtained by lumbar puncture, immediately centrifugated at 1000× *g* at 4 °C for 10 min and aliquoted in polypropylene tubes of 1.5 mL and stored at −80 °C until further analysis, according to international guidelines [47]. AD CSF biomarker measurements were performed in the biochemistry unit of Lariboisière Hospital [36]. Aβ42, Aβ40, t-Tau and p-Tau were first measured by ELISA, using available kits (INNOTEST^®^; Fujirebio, Ghent, Belgium) until July 2018, according to the manufacturer’s instruction. Then, the concentration of these biomarkers was measured by the analyzer Cobas 601 with the electrochemiluminescent reagents Elecsys^®^ (Roche Diagnostics GmbH, Germany). The CSF profiles of patients were classified in A+T+N+/A-T-N-/or intermediate according to cut-offs determined by the laboratory.

CSF levels of α-syn were measured using an ELISA kit (ADx Neuroscience ELISA Alphasynuclein (Ghent, Belgium) available via EUROIMMUN, Lübeck, Germany), validated for research for the quantification of human total α-syn in CSF [48]. 

### 4.5. Ethical Considerations

All the participants provided oral and written information about the opportunity to collect additional blood and CSF samples and perform genetic analyses for further research analyses, according to the BioCogBank© protocol. Written informed consents were obtained for all participants except for two subjects with dementia, under guardianship. Accordingly, consent was obtained from their legal guardian. This study was approved by the local Ethics Committees (*Comité d’évaluation et d’Ethique pour la recherche Paris Nord*) and the *Commission Nationale Informatique et Libertés*.

### 4.6. Statistical Analysis

To describe the study population, continuous variables were presented as means (standard deviations (SD)) and categorical variables were presented as percentages (number of subjects). Analysis of variance (ANOVA) was carried out to compare the CSF concentration of biomarkers between the diagnostic groups, taking into account the stage of cognitive impairment (MCI or dementia). For variables with non-Gaussian distribution, comparisons were made using a Kruskall–Wallis test. Regarding categorical variables, intergroup comparisons were performed using Chi-squared or exact Fisher tests.

We compared the CSF α-syn concentrations with regards to the diagnostic groups and the stage of cognitive impairment, then with regards to their CSF amyloid (A), p-Tau (T) and t-Tau (N) status. Finally, we compared the CSF α-syn concentration given the presence or absence of brain MRI abnormalities. The analysis of Receiver Operating Characteristic (ROC) curve, Se, Spe and AUC were carried out to determine the best α-syn threshold for discriminating subjects with DLB from those with AD or NC. Statistical analyses were performed using R-studio (v1.4.1106).

## 5. Conclusions

Our study demonstrated that α-syn could represent another promising biomarker for improving the discrimination between AD and DLB patients, even at the prodromal stage of the disease. The higher CSF α-syn levels in AD than in DLB may reflect both aggregation of misfolded α-syn in DLB as well as inflammation and faster neurodegeneration in AD. The implementation of CSF α-syn measurement to AT(N) CSF biomarkers could be useful as part of a biomarker panel to support DLB diagnosis. This novel approach could enable an earlier and more accurate diagnosis as well as a better selection of patients for clinical trials of future disease-modifying treatments for DLB. α-syn may also provide additional insights into the prognosis of patients with DLB. Further studies are warranted to precisely determine the role of this biomarker in clinical practice and in therapeutic research.

## Figures and Tables

**Figure 1 ijms-23-13488-f001:**
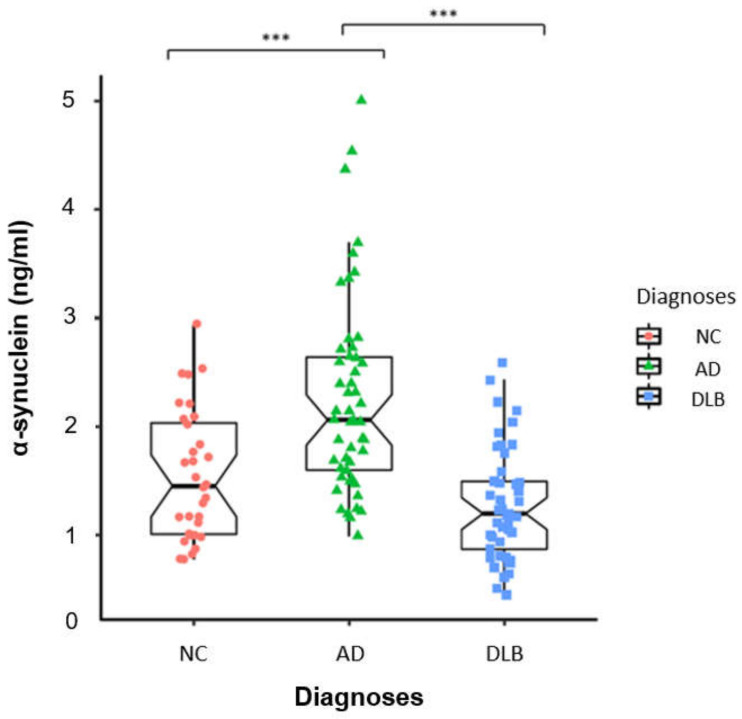
Comparison of the concentration of α-syn between patients with AD, DLB and NC. ***: *p* < 0.001.

**Figure 2 ijms-23-13488-f002:**
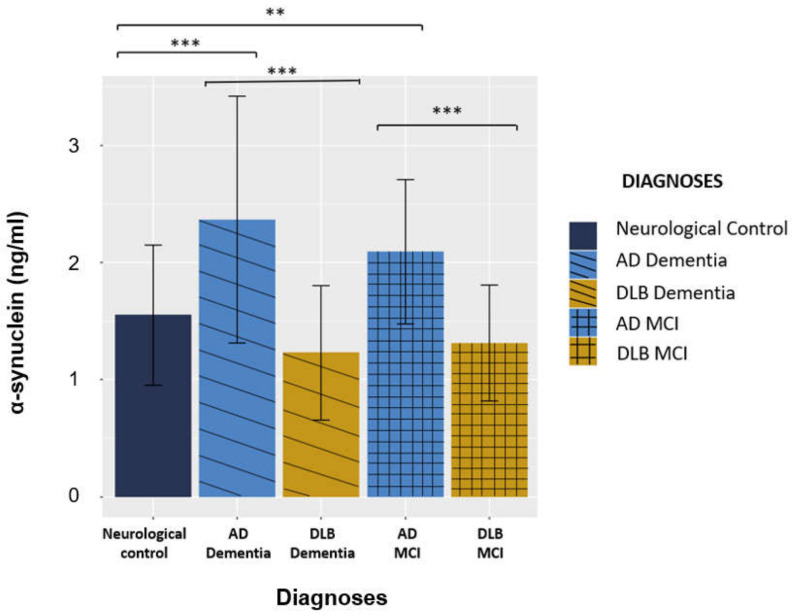
Comparison of the concentration of CSF total α-syn between patients with AD and DLB (MCI or Dementia) and NC. **: *p* < 0.01, ***: *p* < 0.001.

**Figure 3 ijms-23-13488-f003:**
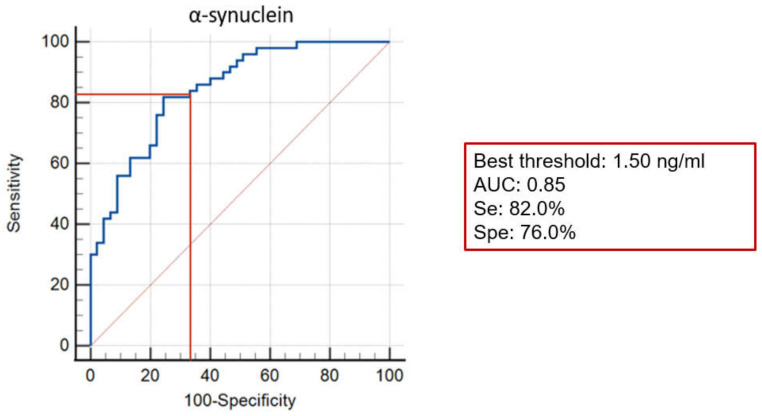
Receiver Operating Characteristic curve: CSF α-syn for discrimination between AD and DLB. AUC: Area Under the Curve, Se: Sensitivity, Spe: Specificity.

**Table 1 ijms-23-13488-t001:** Clinical and biological characteristics of the study population.

Characteristics	Total (*n* = 127)	Diagnostic Groups			
		DLB *n* = 45	AD *n* = 50	NC *n* = 32	*p*-Value
Age (years)	69.3 (8.1)	70.1 (7.0)	71.7 (8.2)	64.4 (7.7)	<0.001
Male gender N (%)	53 (41.7)	28 (62.2)	14 (28.0)	11 (34.3)	<0.01
Disease duration (Y) *	3.0 [1.0–5.0]	3.0 [1.0–5.0]	2.0 [1.0–5.0]	-	0.99
MMSE [0–30] *	24.0 [18.0–27.0]	24.0 [19.0–27.0]	20.0 [15.0–24.8]	27.0 [24.8–28.0]	<0.001
Comorbidities N (%)					
Smoking status	13 (10.2)	5 (11.1)	5 (10.0)	3 (9.4)	1.00
Hypertension	46 (36.2)	15 (33.3)	21 (42.0)	10 (31.3)	0.56
Diabetes	13 (10.2)	3 (6.7)	8 (16.0)	2 (6.3)	0.30
Dyslipidemia	28 (22.0)	6 (13.3)	15 (30.0)	7 (21.9)	0.16
Alcohol	12 (9.4)	6 (13.3)	2 (4.0)	4 (12.5)	0.16
AT(N) biomarkers N (%)					
A +	70 (55.1)	20 (44.4)	50 (100)	0 (0.0)	<0.001
T+	54 (42.5)	4 (8.9)	50 (100)	0 (0.0)	
N+	58 (45.7)	4 (8.9)	50 (100)	4 (12.5)	
MRI Fazekas score N (%)					
0	9 (7.1)	1 (2.2)	3 (6.0)	5 (15.6)	0.16
1–2	67 (52.8)	29 (64.4)	22 (44.0)	16 (50.0)	0.15
3	7 (5.5)	2 (4.4)	5 (10.0)	0 (0.0)	0.07
Missing data	44 (34.6)	13 (29.0)	20 (40.0)	11 (34.4)	0.21
Scheltens score N (%)					
0	10 (5.5)	0 (0.0)	1 (2.0)	9 (28.1)	<0.001
1–2	50 (39.4)	22 (48.9)	16 (32.0)	12 (37.5)	
3–4	21 (16.5)	10 (22.2)	11 (22.2)	0 (0.0)	
Missing data	46 (36.2)	13 (28.9)	22 (44.0)	11 (34.4)	
Microbleeds N (%)					
<5	74 (58.3)	28 (62.2)	25 (50.0)	21 (65.6)	0.17
≥5	1 (0.8)	1 (2.2)	0 (0.0)	0 (0.0)	
Missing data	52 (40.9)	16 (35.6)	25 (50.0)	11 (34.4)	
Brain ischemic injury N (%)	8 (6.3)	5 (11.1)	2 (4.0)	1 (3.1)	0.45
*APOE* ε4 carrier	53 (41.7)	14 (31.1)	31 (62.0)	8 (25.0)	<0.01
Alpha-synuclein (ng/mL)	1.73 (0.83)	1.28 (0.52)	2.26 (0.91)	1.56 (0.60)	<0.001

* Median [1st quartile–3rd quartile].

**Table 2 ijms-23-13488-t002:** A. Comparison of the mean CSF α-syn levels according to the positivity of AT(N) biomarkers. B. CSF α-syn levels regarding white matter lesions on brain MRI. C. α-syn concentrations regarding hippocampal atrophy.

A			
	**CSF α-syn (ng/mL)**		
**CSF Biomarker**	**Negative**	**Positive**	** *p* ** **-Value**
Amyloid (A)	1.49 (0.60)	1.93 (0.95)	**<0.01**
Phosphorylated tau (T)	1.36 (0.54)	2.24 (0.89)	**<0.001**
Neurodegeneration (N)	1.30 (0.49)	2.10 (0.88)	**<0.001**
**B**			
**Fazekas score**	**Mean CSF α-syn (ng/mL)**	**Standard deviation**	** *p* ** **-value**
0	1.87	0.77	0.80
1 or 2	1.70	0.81	
3	1.63	0.58	
**C**			
**Scheltens score**	**Mean CSF α-syn (ng/mL)**	**Standard deviation**	** *p* ** **-value**
0	1.76	0.57	0.89
1 or 2	1.69	0.78	
3 or 4	1.79	0.86	

For detailed cut-offs, see Appendix A.

## Data Availability

The data that support the findings of this study are available from the corresponding author, M.L., upon reasonable request.

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
