# Peer review of "Cerebrospinal Fluid Alpha-Synuclein Improves the Differentiation between Dementia with Lewy Bodies and Alzheimer’s Disease in Clinical Practice"

_ijms, 2022, doi:10.3390/ijms232113488_

Round 1

Reviewer 1 Report

This is an interesting study which aimed to determine and carefully establish the diagnostic value of cerebrospinal alpha-synuclein as a diagnostic biomarker for discriminating between Alzheimer’s disease (AD) and dementia with Lewy bodies (DLB). The authors carefully recruited properly diagnosed patients and measured their lumbar CSF levels of alpha synuclein; and they report lower levels of this biomarker in DLB compared to AD. This difference is attributed to changes in the secretion levels of this protein, but the option of selective death of alpha-synuclein producing neurons as a cause is not considered; the authors may wish to include this option in their discussion.

Another point for considering revision involves the discussion, which is largely dedicated to comparing previous studies by others where the levels of CSF alpha synuclein in the currently discussed diseases were measured and the outcome calculated, in many cases with distinct results from those of the current study. The authors explain their opinion on those previous studies by attributing them to errors in the studied populations or statistical analyses; but the length of this text may irritate some of those authors, and perhaps including these details in a supplementary file may solve this problem.

Author Response

Dear Editor,

Many thanks for your answer, we are delighted to propose our revised version of the manuscript now entitled  “Cerebrospinal fluid alpha-synuclein improves the differentiation between Dementia with Lewy Bodies and Alzheimer’s disease, in clinical practice” (Submission ID ijms-2000921). We have addressed all comments made by the reviewers. The new version of the manuscript has been edited by a professional English proofreading service.

We thank the reviewers for providing guidance that has clearly improved the quality of our manuscript. All page and paragraph numbers refer to locations in the revised manuscript. Changes in the revised version of the manuscript appear in blue.

Responses to Reviewer #1:

Reviewer Comments:

This is an interesting study which aimed to determine and carefully establish the diagnostic value of cerebrospinal alpha-synuclein as a diagnostic biomarker for discriminating between Alzheimer’s disease (AD) and dementia with Lewy bodies (DLB). The authors carefully recruited properly diagnosed patients and measured their lumbar CSF levels of alpha synuclein; and they report lower levels of this biomarker in DLB compared to AD. This difference is attributed to changes in the secretion levels of this protein, but the option of selective death of alpha-synuclein producing neurons as a cause is not considered; the authors may wish to include this option in their discussion.

  • We thank the Reviewer #1 for this suggestion. This very interesting mechanism should have been considered in our manuscript as a causal factor of neurodegeneration. We added two relevant references to explicit this idea 1) Cookson Mol Neurodegeneration 2009 and 2) Osterberg Cell Rep 2015 and modified the first part of the Discussion accordingly P6 L123: “α-syn has been identified as a toxic protein to human neurons and a trigger to neuronal death [21]. More precisely, Lewy inclusion-bearing neurons were shown to selectively die, whereas nonbearing neurons survived [22]. In this context, the distinctive aspects in the pathophysiology of DLB and AD could explain the difference of CSF α-syn levels between these two neurodegenerative disorders”

Another point for considering revision involves the discussion, which is largely dedicated to comparing previous studies by others where the levels of CSF alpha synuclein in the currently discussed diseases were measured and the outcome calculated, in many cases with distinct results from those of the current study. The authors explain their opinion on those previous studies by attributing them to errors in the studied populations or statistical analyses; but the length of this text may irritate some of those authors, and perhaps including these details in a supplementary file may solve this problem.

  • We agree with the Reviewer’s comment and admit that the discussion needed to be reworded concisely. According to this suggestion, we suppressed the paragraph related to the previous studies on this topic and transferred this part in the Supplementary Material. P6 L161 “So far, several studies have examined CSF α-syn in DLB in comparison with AD, other neurodegenerative disorders or NC (for details see Supplementary Material)”.

Reviewer 2 Report

Review of a manuscript “Cerebrospinal fluid alpha-synuclein improves the differentiation between Dementia with Lewy Bodies and Alzheimer’s disease, in clinical practice” by Matthieu Lilamand and coauthors submitted to IJMS.

Dementia with Lewy Bodies and Alzheimer’s disease are prevalent neurological diseases the course of which sometimes is similar and difficult to correctly diagnose. Dementia with Lewy Bodies is often misdiagnosed as Alzheimer’s disease and vice versa and sometimes they may be diagnosed as other cases of synucleinopathies.  The authors aim is to compare α-synuclein level in cerebrospinal fluid (CSF)  of these patients and to study the relationships between the concentration of CSF α-synuclein in comparison with amyloid and tau biomarkers. The authors also compared their data with the neuroimaging characteristics and dementia stages. This is very important biomedical study, the results of which will be interesting for the readers of IJMS.

The following corrections should be made.

Abstract

The more appropriate place for the inclusion criteria for the patients is in the Methods section, but not in Results. Then some additional materials can be added to Results section in Abstract.   

Introduction

Lines 49-50: After the sentence:”DLB is neuropathologically characterized by an accumulation in the brain of abnormally aggregated and phosphorylated alpha-synuclein (α-syn), a 140 amino-acids protein, which is mainly expressed in presynaptic terminals” the authors should add a reference on the following review: “Synucleins and Gene Expression: Ramblers in a Crowd or Cops Regulating Traffic? Front Mol Neurosci. 2017 Jul 13;10:224. doi: 10.3389/fnmol.2017.00224.”

Lines 51-52:”This accumulation is often associated with brain amyloid deposition and both participate in neurodegeneration [5].” This is unclear sentence. Of the authors mean by “both” amyloid and a-synuclein, they should clearly say so.   

Results

Line 73: ”we also randomly included 50 patients with biologically confirmed AD.” The authors should explain more clearly what they mean by “biologically confirmed”

Figure 1 It is unclear why the a-synuclein level is expressed in pg/ml, but not in ng/ml. This change would simplify Y-axis and allow to get rid of many zeros.

All Figure legends : there is no need to give full names of all abbreviations, i.e. MCI, NC, AD, DLB, since it was done many times before.

Discussion

Lines 149-150: “As a result, both aggregates of α-syn in DLB and overexpression of wild-type α-syn in AD favor synaptotoxicity.” Do the authors mean that overexpression of wild-type α-syn in AD is not accompanied with its aggregation?

Lines 201-202: ”Current new technology (e.g. Real-Time Quaking-Induced Conversion) are being developed to measure abnormal α-syn, but their usefulness in daily clinical practice of laboratories deserves further investigation.” The authors should correct the sentence and add the citation as follows: ”Current new technologies (e.g. Real-Time Quaking-Induced Conversion and Protein Misfolding Cyclic Amplification [PMCA]) are being developed to measure abnormal α-syn, but their usefulness in daily clinical practice of laboratories deserves further investigation”; citation: “Analysis of Protein Conformational Strains-A Key for New Diagnostic Methods of Human Diseases. Int J Mol Sci. 2020 Apr 17;21(8):2801. doi: 10.3390/ijms21082801.”

Author Response

Dear Editor,

Many thanks for your answer, we are delighted to propose our revised version of the manuscript now entitled  “Cerebrospinal fluid alpha-synuclein improves the differentiation between Dementia with Lewy Bodies and Alzheimer’s disease, in clinical practice” (Submission ID ijms-2000921). We have addressed all comments made by the reviewers. The new version of the manuscript has been edited by a professional English proofreading service.

We thank the reviewers for providing guidance that has clearly improved the quality of our manuscript. All page and paragraph numbers refer to locations in the revised manuscript. Changes in the revised version of the manuscript appear in blue.

Responses to Reviewer #2:

Reviewer Comments:

Review of a manuscript “Cerebrospinal fluid alpha-synuclein improves the differentiation between Dementia with Lewy Bodies and Alzheimer’s disease, in clinical practice” by Matthieu Lilamand and coauthors submitted to IJMS.

Dementia with Lewy Bodies and Alzheimer’s disease are prevalent neurological diseases the course of which sometimes is similar and difficult to correctly diagnose. Dementia with Lewy Bodies is often misdiagnosed as Alzheimer’s disease and vice versa and sometimes they may be diagnosed as other cases of synucleinopathies.  The authors aim is to compare α-synuclein level in cerebrospinal fluid (CSF)  of these patients and to study the relationships between the concentration of CSF α-synuclein in comparison with amyloid and tau biomarkers. The authors also compared their data with the neuroimaging characteristics and dementia stages. This is very important biomedical study, the results of which will be interesting for the readers of IJMS.

The following corrections should be made.

Abstract

The more appropriate place for the inclusion criteria for the patients is in the Methods section, but not in Results. Then some additional materials can be added to Results section in Abstract.  

  • Consistent with the Reviewer’s suggestion, we suppressed the number of the included patients with AD, DLB and NC from the Results of the Abstract. Instead, we added the following information L26 “without differences due to the stage of cognitive impairment”

Introduction

Lines 49-50: After the sentence:”DLB is neuropathologically characterized by an accumulation in the brain of abnormally aggregated and phosphorylated alpha-synuclein (α-syn), a 140 amino-acids protein, which is mainly expressed in presynaptic terminals” the authors should add a reference on the following review: “Synucleins and Gene Expression: Ramblers in a Crowd or Cops Regulating Traffic? Front Mol Neurosci. 2017 Jul 13;10:224. doi: 10.3389/fnmol.2017.00224.”

  • We thank the Reviewer for this suggestion and added the mentioned reference in the revised version of the manuscript.

Lines 51-52:”This accumulation is often associated with brain amyloid deposition and both participate in neurodegeneration [5].” This is unclear sentence. Of the authors mean by “both” amyloid and a-synuclein, they should clearly say so.  

  • We apologize for this inaccuracy. Consistent with the Reviewer’s advice we stated P2 L52 that “both amyloid and α-syn participate in neurodegeneration”.

Results

Line 73: ”we also randomly included 50 patients with biologically confirmed AD.” The authors should explain more clearly what they mean by “biologically confirmed”

  • We acknowledge that the term biogically confirmed could have been confusing, especially when the Methods are detailed after the Results section. For clarity purpose:
    • L74 We replaced “biologically confirmed” by “AD confirmed by CSF biomarkers”
    • In the Methods P7 L222 we stated that “In parallel, we randomly included the same number of patients with AD meeting the clinical criteria of Mc Khann’s criteria [41] and confirmed by CSF biomarkers (i.e. with CSF A+T+N+ profile). These individuals with AD and available additional CSF samples, were included over the same inclusion period, and were individually matched for age and gender with the patients with DLB”.

Figure 1 It is unclear why the a-synuclein level is expressed in pg/ml, but not in ng/ml. This change would simplify Y-axis and allow to get rid of many zeros.

  • We thank the Reviewer for raising this issue. We now express alpha synuclein levels in ng/ml throughout all the revised version of the manuscript, including the Tables and Figures.

All Figure legends : there is no need to give full names of all abbreviations, i.e. MCI, NC, AD, DLB, since it was done many times before.

  • These abbreviations have been deleted, as suggested by the Reviewer.

Discussion

Lines 149-150: “As a result, both aggregates of α-syn in DLB and overexpression of wild-type α-syn in AD favor synaptotoxicity.” Do the authors mean that overexpression of wild-type α-syn in AD is not accompanied with its aggregation?

  • We thank the Reviewer for this very interesting comment. The pathophysiology of α-syn in AD is very complex and still debated (Example: Twohid D. Molecular Neurodegeneration 2019). Neuropathological studies reported that patients with AD show Lewy-related pathology due to α-syn aggregation. However, this mechanism seems less significant in AD than the overexpression α-syn, which triggers toxic effects as well as elevated CSF levels of α-syn. That is why we presented on the one hand the aggregation mechanism which is paramount in DLB (and PD) and on the other hand the overexpression of α-syn.in AD patients.

Lines 201-202: ”Current new technology (e.g. Real-Time Quaking-Induced Conversion) are being developed to measure abnormal α-syn, but their usefulness in daily clinical practice of laboratories deserves further investigation.” The authors should correct the sentence and add the citation as follows: ”Current new technologies (e.g. Real-Time Quaking-Induced Conversion and Protein Misfolding Cyclic Amplification [PMCA]) are being developed to measure abnormal α-syn, but their usefulness in daily clinical practice of laboratories deserves further investigation”; citation: “Analysis of Protein Conformational Strains-A Key for New Diagnostic Methods of Human Diseases. Int J Mol Sci. 2020 Apr 17;21(8):2801. doi: 10.3390/ijms21082801.”

  • We thank very much the Reviewer for this insightful remark. The correction has been made P7 L221 and this relevant reference has been added.